# Molecular understanding of polyelectrolyte binders that actively regulate ion transport in sulfur cathodes

Longjun Li [1], Tod A. Pascal [2], Justin G. Connell [3], Frank Y. Fan[4], Stephen M. Meckler[5], Lin Ma[1], Yet-Ming Chiang[4], David Prendergast [1,2] & Brett A. Helms [1,2]

Polymer binders in battery electrodes may be either active or passive. This distinction depends on whether the polymer influences charge or mass transport in the electrode. Although it is desirable to understand how to tailor the macromolecular design of a polymer to play a passive or active role, design rules are still lacking, as is a framework to assess the divergence in such behaviors. Here, we reveal the molecular-level underpinnings that distinguish an active polyelectrolyte binder designed for lithium–sulfur batteries from a passive alternative. The binder, a cationic polyelectrolyte, is shown to both facilitate lithium-ion transport through its reconfigurable network of mobile anions and restrict polysulfide diffusion from mesoporous carbon hosts by anion metathesis, which we show is selective for higher oligomers. These attributes allow cells to be operated for >100 cycles with excellent rate capability using cathodes with areal sulfur loadings up to 8.1 mg cm$^{-2}$.

[1] The Joint Center for Energy Storage Research, Lawrence Berkeley National Laboratory, Berkeley, CA 94720, USA. [2] The Molecular Foundry, Lawrence Berkeley National Laboratory, 1 Cyclotron Road, Berkeley, CA 94720, USA. [3] The Joint Center for Energy Storage Research, Argonne National Laboratory, Argonne, Lemont, IL 60439, USA. [4] Department of Materials Science and Engineering, Massachusetts Institute of Technology, Cambridge, MA 02139, USA. [5] Department of Chemistry, University of California, Berkeley, CA 94720, USA. Correspondence and requests for materials should be addressed to B.A.H. (email: bahelms@lbl.gov)

Active layers in electrochemical energy storage devices typically incorporate polymer binders to aid in processing composite electrodes with well-controlled architecture and compliant mechanical integrity. Polymer binders also dictate the extent of electrode swelling with electrolyte and help mitigate cracking on drying or swelling, or on large volume changes experienced using certain electrode chemistries between their extremes in state-of-charge[1–4]. Often overlooked is whether a polymer binder is an active or a passive component in the composite electrode, a distinction that denotes whether or not it participates in charge or mass transport; it can also be adaptive if it can be made to switch between passive and active states, e.g., using thermal excursions or redox chemistry[5–7]. Whereas the chemical constitution of a polymer binder should dictate whether it is passive, active, or adaptive in the electrode, it remains a challenge to reveal the molecular basis by which these behaviors manifest. Without this information, rational design principles for polymer binders remain obscure.

Here, we elucidate the molecular-level underpinnings that distinguish an active polymer binder designed for the lithium–sulfur (Li–S) battery from a ubiquitous yet passive alternative (Fig. 1). The macromolecular structure of our polyelectrolyte binder—poly[(N,N-diallyl-N,N-dimethylammonium) bis(trifluoromethanesulfonyl)imide] (PEB-1)—allows two types of ion transport critical to the operation of a Li–S cell to be actively facilitated or regulated: facilitated transport of lithium ions throughout the sulfur cathode, which manifests as low charge-transfer resistance and fast electrode kinetics; and restricted diffusion of soluble polysulfides ($Li_2S_n$, where $n = 4$–8)

from nitrogen-doped mesoporous carbon (N-MC) sulfur hosts into the electrolyte. The emerging perspective from our work is that the design space for polyelectrolyte binders is superior, allowing for broad tunability of electrochemical stability (both anodic and cathodic), energetic barriers to $Li^+$ desolvation and transport, and adhesion.

Our success highlights the profound yet often underappreciated importance of macromolecular design and mechanistic understanding of active polymer binders in Li–S battery technology development, where the role of the binder in the cathode should be considered in stride with other cell components, including sulfur-rich polymers[8–11], sulfur host materials[12–15], embedded current collectors[16–19], separators[20,21], redox mediators[22–24], electrolytes[25–30], and ionically conductive surface films for long-term lithium metal protection[23,31–34]. If successful, this battery chemistry is well-positioned to augment the electrochemical energy storage options for transportation, aviation, and light-weight portable power[35,36]—and may ultimately be the most sustainable solution for these applications given the prevalence and low cost of sulfur relative to transition metals used in conventional Li-ion intercalation solids[37–39]. Our results are complementary to advances in Li-ion battery technology development using polyelectrolyte binders (e.g., poly(ionic liquid)s), which yielded cells with high specific capacity and excellent long-term electrochemical stability when compared to PVDF binder[40–42].

Notably, we demonstrate in the context of Li–S cells that PEB-1 binders make possible their operation with high accessible areal capacity (e.g., up to 8.13 mAh cm$^{-2}$) and excellent rate capability

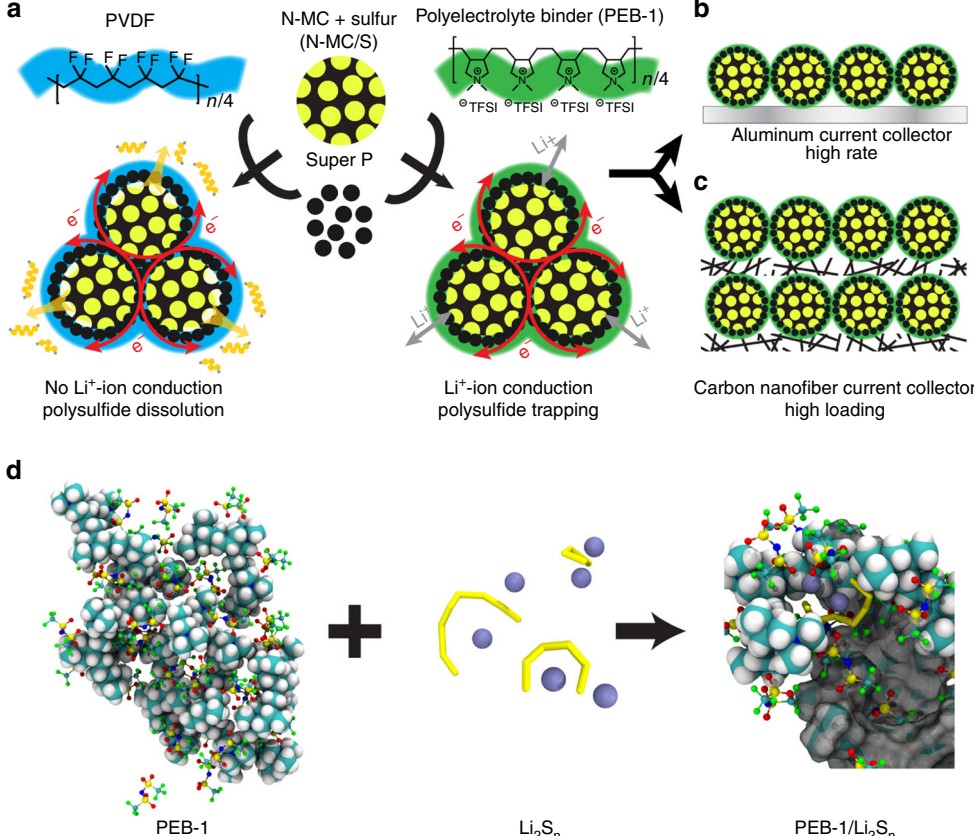

**Fig. 1** Illustration of the fabrication of sulfur electrodes with PVDF or PEB-1 binder. **a** The cathode is comprised of sulfur-active materials loaded into N-doped mesoporous carbon (N-MC) hosts, 'Super P' as the conductive additive, and a polymer binder (PEB-1 or PVDF). **b** A conventional sulfur cathode cast onto an aluminum current collector. **c** A highly loaded sulfur cathode cast onto a carbon nanofibre current collector. **d** Schematic illustrating the formation of complex ion clusters via anion metathesis, when PEB-1 encounters soluble polysulfides during Li–S cell cycling

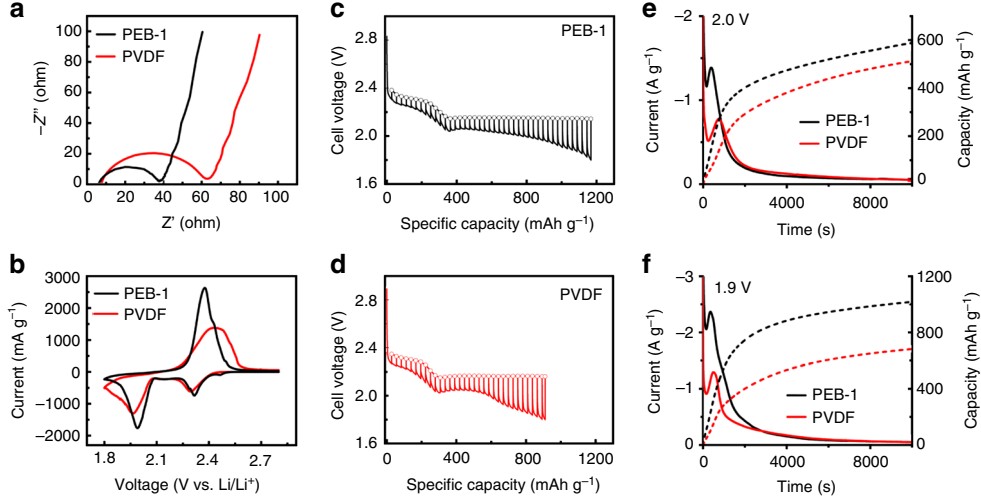

**Fig. 2** Improved cell kinetics enabled by PEB-1 compared with conventional PVDF binder. **a** Electrochemical impedance spectroscopy (EIS) for Li–S cells after cell assembly. **b** Cyclic voltammetry (CV) for Li–S cells after cell assembly. **c**, **d** Galvanostatic intermittent titration technique (GITT) for Li–S cells on the first discharge. **e**, **f** Potentiostatic discharge experiments for Li–S cells on the first discharge, after being equilibrated at 2.3 V for 6 h. The cell voltage was then lowered to either 2.0 V or 1.9 V to initiate the nucleation and growth of Li₂S on the embedded current collector

(e.g., C/5–2C) using cathodes with high-areal sulfur loadings (e.g., up to 8.1 mg cm$^{-2}$). On the other hand, composite sulfur cathodes prepared using poly(vinylidene difluoride) (PVDF), a common but passive polymer binder, exhibit slower and rapidly degrading electrode kinetics. In turn, PVDF cells access lower capacity and experience shorter cycle life.

## Results

**Designing polyelectrolyte binders to facilitate Li-ion transport.** Facilitated transport of lithium ions in a composite electrode is critical to enabling high-rate chemical transformations with sulfur-active materials[43,44]. This concerns both long-range ion and mass transport within the pore voids of the cathode to reduce electrode polarization, and short-range ionic charge transfer across the electrolyte–sulfur host interface (e.g., the orifice of a mesopore containing sulfur) to ensure fast sulfur interconversion kinetics. PEB-1 was designed to achieve this by allowing Li-ion hopping along a plurality of weakly associated, and thus mobile, bis(trifluoromethanesulfonyl)imide (TFSI$^-$) counterions associated with the cationic polymer backbone[45]. In the past, alternative PEBs aiming to facilitate Li-ion transport along anionic moieties bound to the polymer backbone—including lithiated carboxymethylcellulose/styrene butadiene rubber blends[2], lithiated Nafion/polyvinylpyrollidinone blends[46], or sulfonated poly (ether ether ketone)[47]—have been implemented with limited success, likely owing to ion clustering during cathode processing, which prevents the ion-transporting polymer domains from percolating effectively throughout the cathode[48–51]. As such, they have not been used in Li–S cells pushing the limits of energy density and power, as is demonstrated here using PEB-1.

To confirm that the macromolecular design of PEB-1 improves the sulfur cathode's electrochemically driven S$_8$/Li$_2$S interconversion kinetics, electrochemical impedance spectroscopy (EIS), cyclic voltammetry (CV), galvanostatic intermittent titration technique (GITT), and potentiostatic discharge experiments were carried out on Li–S cells prepared with either PEB-1 or PVDF as the binder. We found that the EIS profiles for each cell type exhibited a high-frequency intercept on the real axis (ohmic resistance, or $R_s$), a medium-frequency semicircle (charge-transfer resistance, or $R_{ct}$), and a low-frequency incline (Warburg impedance, or $Z_w$) (Fig. 2a). Cells incorporating PEB-1 exhibited

both smaller $R_s$ and $R_{ct}$, indicating larger active-carbon surface area and better electronic and ionic conductivity. These attributes stand in contrast to the ionically insulating and pore-blocking character of PVDF at the electrolyte–mesopore interface of N-MC hosts for sulfur-active materials[3,5].

Although EIS provides static evidence for reduced cell impedance with PEB-1 in place, deeper insight into how PEB-1 influences sulfur redox reactions within N-MC hosts from a dynamic perspective is gleaned from CV and GITT experiments. Comparing cyclic voltammograms for PEB-1 and PVDF-based Li–S cells (Fig. 2b), PEB-1 cells showed higher peak current densities and considerably smaller overpotentials for both reduction and oxidation. As there is no contribution to capacity from PEB-1 (Supplementary Fig. 1), performance gains observable in the CV are due to increased sulfur utilization and faster reaction kinetics. We further show using GITT experiments (Fig. 2c, d) that electrochemical utilization of sulfur is more efficient with PEB-1 than PVDF binder in the cells. At a discharge rate of C/5, the overall discharge capacity of PEB-1 cell was 257 mAh g$^{-1}$ more than that of PVDF cell, revealing there was 28% more sulfur reduced in PEB-1 cells during the initial discharge process. In the upper discharge voltage plateau, both the quasi-equilibrium potentials and the discharge potentials of PEB-1 cells are higher than those of PVDF cells, indicating there is less barrier for Li$^+$ ions to cross the binder-sulfur interface and react with sulfur and a higher concentration of LiTFSI at the reaction front (generated by anion metathesis). In the lower discharge voltage plateau, the quasi-equilibrium potentials are almost the same in both cells owing to a liquid–solid phase transition between Li$_2$S$_4$ and Li$_2$S$_2$/Li$_2$S and completion of anion metathesis[52]. However, the discharge potentials of PEB-1 cells are still higher than PVDF cells owing to the lower barrier for Li$^+$ ion transport in the dynamic process, leading to higher overall specific capacity. There is some evidence that LiTFSI oxidizes sulfur compounds to higher oxidation states, which would contribute to capacity fade[53,54]. However, based on the superior electrochemical performance of PEB-1 cells, this effect does not appear to factor strongly here[28].

The lower overpotential for sulfur reduction on discharge strongly influenced the rate of Li$_2$S electrodeposition on the conductive carbons, which was quantified using potentiostatic discharge experiments. Here, all S$_8$ and higher order polysulfides

were initially reduced to $Li_2S_4$ (nominally) by holding the cells at 2.3 V for 6 h, after which the current became negligible. The cell voltage was then lowered to either 2.0 V or 1.9 V to initiate the nucleation and growth of $Li_2S$, and the current transient was monitored as the system reached steady state (Fig. 2e, f). In both cases, there is a peak in the current transient due to the nucleation of $Li_2S$, and a tail owing to the consumption of available polysulfides and passivation of carbon surface. The peak in the current transient for PEB-1 cells was significantly earlier compared to PVDF cells; furthermore, the peak height was higher, yielding higher capacity delivered at steady state.

To understand the relative contributions of double layer charging, reduction of higher order polysulfides, or electrodeposition on the observed current transients, we carried out mathematical modeling of 2D $Li_2S$ growth on the conductive carbon surface. We used the following relationship (Eq. 1) between current density ($J$) and reaction time ($t$)[22,55]:

$$\frac{J}{J_m} = \left(\frac{t}{t_m}\right) \exp\left[-\frac{1}{2}\left(\frac{t^2}{t_m^2} - 1\right)\right] \quad (1)$$

where, $J_m$ and $t_m$ represent the maximum current density and the time it occurs, respectively. Based on the fitting results, the rate constant ($\kappa$) of lateral film growth for $Li_2S$ electrodeposits was calculated using Eq. 2:

$$t_m = \left(2\pi N_0 \kappa^2\right)^{-\frac{1}{2}} \quad (2)$$

where, $N_0$ represents the areal density of $Li_2S$ nuclei, and $N_0 \kappa^2$ gives the effective rate constant for the coverage of carbon surface. At 2.0 V, the effective rate constant enabled PEB-1 is $7.84 \times 10^{-7}$ s$^{-2}$, which is more than two times higher than that observed using PVDF ($3.18 \times 10^{-7}$ s$^{-2}$). The coverage of the carbon surface is accelerated by PEB-1 owing to: (1) facilitated Li-ion conductivity providing faster mass transport; (2) polysulfide trapping by PEB-1 keeps polysulfides close to reaction sites.

**Designing polyelectrolyte binders to regulate polysulfide transport.** In addition to the importance of facilitated Li-ion transport on Li–S cell impedance and cathode reaction kinetics, regulated transport of polysulfides is key to capacity retention, Coulombic efficiency, and long-term Li-anode stability, particularly at high-areal sulfur loadings[56]. Such regulation was also considered in the design of PEB-1, where we sought to leverage the anion metathesis between polymer-associated TFSI$^-$ anions and the anionic end-groups of lithium polysulfides that are generated during intermediate states-of-charge of the Li–S cell.

Specifically, $LiS_n$ radical anions could bind at their anionic terminus to cationic pyrrolidinium polymer moieties; on the other hand, $Li_2S_n$ dianionic polysulfides can bind through either a single terminus or both termini. In the case of the latter, $Li_2S_n$ can bind to the same polymer chain twice (i.e., an intramolecular crosslink), or to two adjacent polymer chains (i.e., an intermolecular crosslink). For both, this requires the initially ring-closed $Li_2S_n$ species to exchange its most labile lithium ion for the pyrrolidinium cation prior to ring-opening and exchange of the second lithium ion with another pyrrolidinium cation (Fig. 1d)[57,58]. Any of the binding modes described is concomitant with the generation of either one or two molecular equivalents of LiTFSI, which aids in maintaining high [LiTFSI] where polysulfides are generated, as needed for fast electrode kinetics.

The excellent polysulfide-binding character of PEB-1 (Supplementary Fig. 2a) was initially demonstrated by introducing PEB-1 solids to solutions of lithium polysulfides (10 mM, prepared as $Li_2S_6$) in a 1:1 $v/v$ mixture 1,3-dioxolane/1,2-dimethoxyethane (DOL/DME) containing LiTFSI (1.0 M) and LiNO$_3$ (0.20 M), the

electrolyte used in our Li–S cell; PEB-1 is not soluble in DOL: DME electrolyte. After 1 h, 96% of the polysulfides had been leached from solution through anion metathesis, as determined by optical spectroscopy of the filtrate (Supplementary Fig. 3)[59]. The PEB-1/polysulfide composite (Supplementary Fig. 2a) was isolated by filtration and dried under inert conditions before further analysis by X-Ray photoelectron spectroscopy (XPS) and X-Ray absorption spectroscopy (XAS). A similar phenomenon could be observed without LiTFSI and LiNO$_3$ in the electrolyte (Supplementary Fig. 2b), illustrating the strong absorption behavior of PEB-1, even without supporting salts.

The extent of anion metathesis was determined by XPS, taking advantage of the unique chemical signatures in the $N$ 1 s spectra for pyrrolidinium and TFSI$^-$ moieties of PEB-1 (Fig. 3a). The peak at 402.8 eV is assigned to the quaternary nitrogen atom in the pyrrolidinium ring (i.e., $N_{cation}$), whereas the peak with lower binding energy at 399.2 eV is assigned to the nitrogen atom in TFSI$^-$ (i.e., $N_{anion}$). The peak area ratio between $N_{cation}$ and $N_{anion}$ is ~1:1, as is expected from the chemical structure of PEB-1. However, as shown in Fig. 3b, anion metathesis results in a near-complete loss of $N_{anion}$ at 399.2 eV, indicating replacement of TFSI$^-$ anions with $S_n^{2-}$ anions. These peak assignments were validated by density functional theory (DFT) calculations of the valence electron charge density around the selected atom (Supplementary Fig. 4). Here, the lower valence electron density around the excited atom leads to higher XPS-binding energy, and vice versa.

Although lithium polysulfides in electrolyte are mixtures of $LiS_n$ and $Li_2S_n$, where $n$ varies by states-of charge, $Li_2S_n$ are the most prevalent[58]. We were then interested in understanding the binding geometries available to $Li_2S_n$ for different oligomer length $n$ to interact with one or more cationic polymer chains. We accomplished this by means of semi-classical, accelerated molecular dynamics simulations. Model systems, comprising a short oligomer of PEB-1 (in this case a tetramer), along with its TFSI$^-$ counterions, and various oligomer lengths of $Li_2S_n$ were considered, and the binding free energy determined from extensive metadynamics simulations[60,61]. Critically, we employed a fluctuating charge model to facilitate intermolecular charge transfer and intramolecular charge reorganization[62,63], which we found necessary to accurately describe the physics in these systems. Details of our computational methodology are given in the Supplementary Methods.

At equilibrium, TFSI$^-$ and $Li_2S_n$ molecules both preferentially interact with pyrrolidinium subunits of the polymer, resulting in ion clusters. We find that the binding free energy of an isolated $Li_2S_n$ molecule to PEB-1 increases with increasing polysulfide chain length, ranging from −48 kJ mol$^{-1}$ for $Li_2S_4$ to −69 kJ mol$^{-1}$ for $Li_2S_8$ (Fig. 3e, Supplementary Fig. 5a). Generally, we find that the binding free energy of TFSI$^-$ to PEB-1 is less than the lithium polysulfides, ranging from −52 kJ mol$^{-1}$ with no lithium polysulfides present, −30 kJ mol$^{-1}$ with $Li_2S_8$ present, and −40 kJ mol$^{-1}$ with $Li_2S_4$ present (Fig. 3f, Supplementary Fig. 5b). Considering the overall equation $Li_2S_n$ + TFSI-PEB ->LiTFSI + $LiS_n$-PEB, the driving force for metastasis is the ease of formation of LiTFSI. For shorter chain polysulfides, the lithium cations are tightly bound, owing to the large electrostatic interactions resulting from the more polar polysulfide molecules. Indeed, the equilibrium binding geometry is symmetric, with the lithium ions equidistant above and below the polysulfide plane[58]. The reduction in the average partial atomic charge in the larger polysulfides leads to an equilibrium binding geometry comprising a tightly bound and a more labile lithium ion in a ring-closed polysulfide. The ease of extracting the more labile polysulfide increases with increasing polysulfide chain length, thus promoting metathesis in an oligomer-selective manner. Upon anion metathesis, the nominal

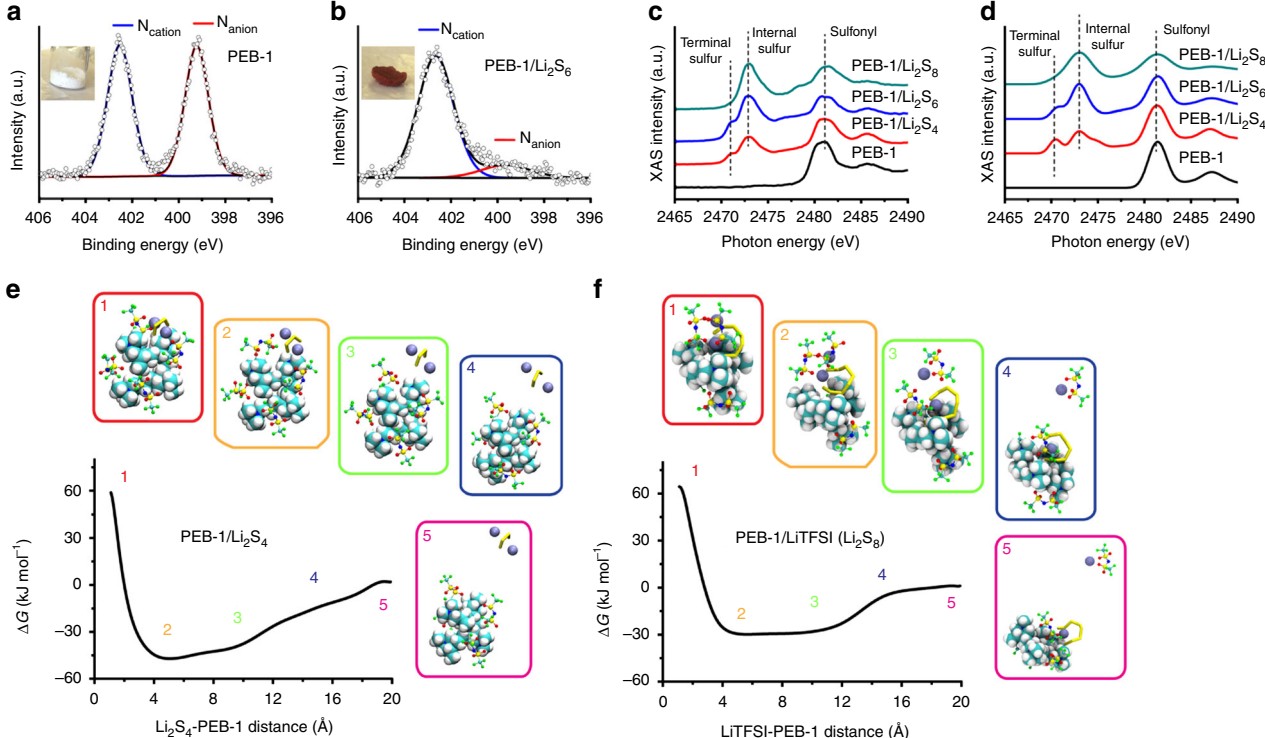

**Fig. 3** Analysis of the polysulfide–PEB-1 composite. **a, b** X-ray photoelectron spectroscopy (XPS) of the $N$ 1 s signal. **c** Sulfur K-edge X-ray absorption (XAS) spectra. **d** Calculated XAS spectra. **e** Free energy corresponding to the interaction of PEB-1 and $Li_2S_4$ when LiTFSI is present in the ion cluster. **f** Free energy corresponding to the interaction of PEB-1 and LiTFSI when $Li_2S_8$ is present in the ion cluster

ring-closed $LiS_n$ ring-opens and undergoes a second metathesis event, leading to either inter- or intramolecular crosslinks. The longer oligomers were more effective at intermolecular cross-linking in the simulations (Fig. 1d).

Predictions suggesting PEB-1 preferentially binds to higher order polysulfides were confirmed by S K-edge XAS[57]. As noted above, the terminal atoms on the polysulfide chain are more negatively charged than the bridging sulfur atoms. Spectroscopically, this manifests as a pre-edge feature near 2471 eV in the sulfur K-edge XAS, distinct from the regular sulfur "white-line" peak near 2472 eV. Further, theoretical calculations have shown that the relative intensity of this peak is a function of the ratio of the number of bridging/terminal sulfur atoms, such that $Li_2S_4$ has a more pronounced pre-peak than $Li_2S_8$. This insight has been used to fit experimental XAS data[64,65] to determine speciation in working Li–S cells. Alternative approaches have utilized experimentally derived standards of related ionic crystals to determine speciation[66]. PEB-1 shows two major peaks around 2481 and 2486 eV, which are contributed by the sulfonyl functional group in TFSI[-] anions (Fig. 3c). For PEB-1/$Li_2S_8$, PEB-1/$Li_2S_6$, and PEB-1/$Li_2S_4$, we observe an additional peak and its shoulder at 2473 eV and 2471 eV, which are due to, respectively, bridging and terminal sulfur atoms in polysulfides[67]. As the oligomer length of $Li_2S_n$ increases, the peak at 2473 eV also increases, which is due to the presence of more sulfur in the PEB-1/$Li_2S_n$ composite. The peak area ratio of the terminal and bridging sulfur atoms decreases as $n$ increases from 4 to 6 and 8, which is in agreement the theoretical composition of $S_8^{2-}$, $S_6^{2-}$, and $S_4^{2-}$. The experimental XAS absorption spectra are in agreement with DFT calculations (Fig. 3d). Of interest is the XAS spectrum of the PEB-1/$Li_2S_8$ complex. We would expect a significant, though reduced, pre-edge feature for the ring-closed structure, consistent with the XAS of the shorter chained polysulfides. However, the experimental XAS spectrum shows no evidence of a pre-edge

feature, which is only recovered computationally when considering a ring-opened structure (Supplementary Fig. 6), where the terminal sulfur atoms are more covalently bonded to the pyrrolidinium nitrogen atoms that pull electron density away.

Conventional anionic (and lithiated) polyelectrolyte binders are incapable of such cross-linking with polysulfides, and thus do not actively regulate polysulfide migration; PEB-1 is unique in that regard. Ultimately, we view PEB-1's role in regulating polysulfide migration as analogous to a variety of inorganic adsorbents recently described by Cui[68], Nazar[69], and others[70,71] with the added benefits of facilitating Li-ion transport and also serving as a binder. PEB-1 may also find more practical use than other cationic binders for Li–S cells, e.g., poly(acrylamide-*co*-diallyldimethylammonium chloride), which corrodes Al current collectors[72]. By obviating the use of corrosive Cl[-] counterions, PEB-1 is a preferred embodiment. It may also be the case that PEB-1 influences the electrolyte system within the electrode's pores, altering the chemistry and solvated structures of polysulfides and the working ion in the pores. This has been suggested for binders like poly(ethylene oxide)[73,74], which delays the passivation of the cathode at end of discharge rather than a trapping capacity for polysulfides[73].

**Incorporating active PEB-1 binders in composite sulfur cathodes.** Alongside the polymer binder and a highly networked current collector (e.g., Super P, carbon nanofibres (CNFs), or multiwalled carbon nanotubes), sulfur is typically introduced to the cathode as $S_8$ that has been encapsulated in a porous host. Here, we melt-infused commercial sulfur powders into a nitrogen-doped mesoporous carbon (N-MC/S), whose surface area was 677 $m^2 g^{-1}$ (Supplementary Fig. 7) and whose nitrogen content was 12% w/w (Supplementary Fig. 8) with respect to carbon; the sulfur loading in the N-MC host was ~ 80% w/w, as determined by thermogravimetric analysis (TGA, Supplementary

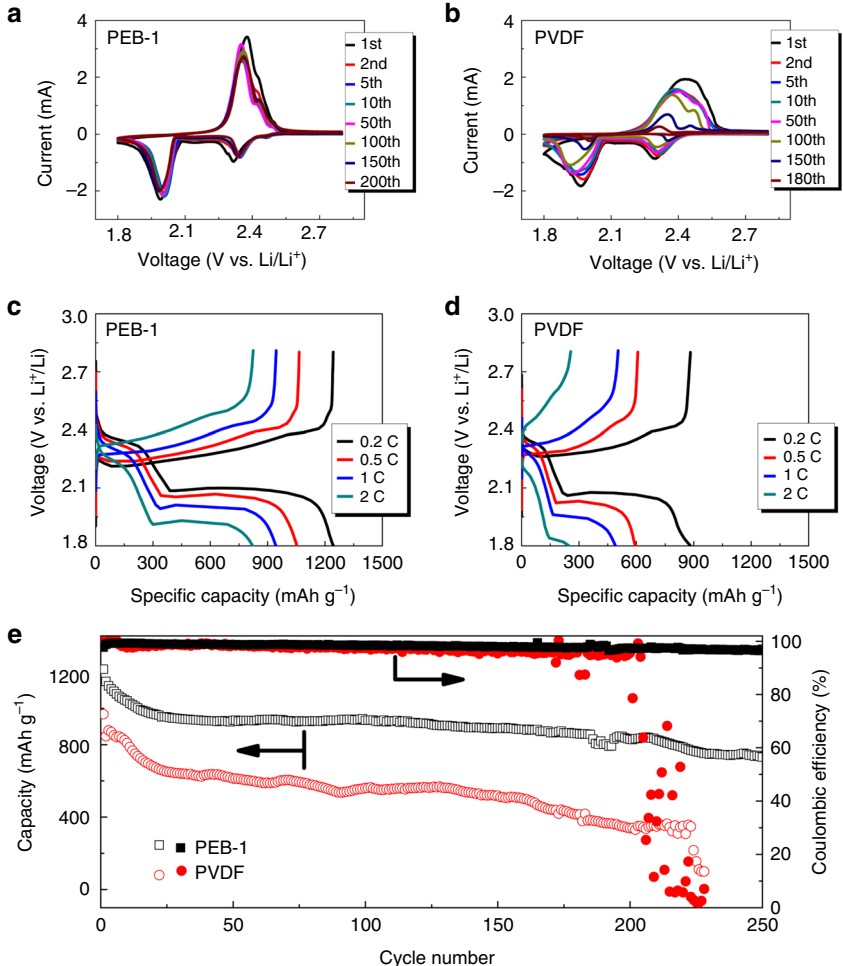

**Fig. 4** Electrochemical performance of Li–S full cells with PEB-1 or PVDF binder. **a**, **b** Long-term cyclic voltammetry (CV). **c**, **d** Discharge and charge curves at different C rates. **e** Cycling performance for each cell type at a rate of C/5. In all cases, the composite sulfur electrode was cast onto an aluminum current collector

Fig. 9). Owing to its architecture, N-MC retains most sulfur inside the carbon particles, which have a high internal to external surface area ratio. Nitrogen-doping promotes the chemical adsorption between sulfur atoms and N-MC[13]. In this way, polysulfides near the orifice of a mesopore are most subject to diffusion and loss. However, when N-MC/S is coated with PEB-1 during electrode processing, polysulfides diffusion can be effectively managed. Toward that end, slurries were then prepared by combining N-MC/S, Super P, and either PVDF or PEB-1 in a mass ratio of 7:2:1 in NMP. For low areal sulfur loadings, slurries were cast onto a conventional aluminum current collector (Al-CC) and dried at 50 °C for 12 h (Fig. 1b). SEM images of PEB-1 and PVDF electrodes (Supplementary Fig. 10) show few macroscopic structural differences with respect to the distribution of N-MC/S composite and Super P particles (neither binder is directly observable). This is not too surprising as the binder loading is the same for both electrode formulations, and the surface area of N-MC/S is small (5.2 m$^2$ g$^{-1}$, see Supplementary Fig. 7). For areal sulfur loadings exceeding ~ 2 mg cm$^{-2}$, we found it necessary to implement a CNF paper as the current collector (CNF-CC, Fig. 1c) in place of aluminum. Initially described by Manthiram and co-workers[17], the web-like architecture of the CNF-CC better accommodates volume changes on slurry drying, particularly for large volumes of viscous slurries, yielding a mechanically robust electrode suitable for Li–S cell assembly. The areal loading of sulfur in the CNF-CC was

~ 4 mg cm$^{-2}$. High overall sulfur loadings, up to ~ 8 mg cm$^{-2}$, simply required the use of two coated CNF-CCs in a layered architecture.

**Li–S cell performance gains when active PEB-1 binders are in place**. To confirm the positive impact of PEB-1 in Li–S cells, we assembled and tested coin cells with either PEB-1 or PVDF binder (denoted hereafter as PEB-1 or PVDF cells). CV tests were carried out between 1.8 V and 2.8 V vs. Li/Li$^+$ at a scan rate of 0.1 mV s$^{-1}$. As shown in Fig. 4a, b, both CVs possess two major reduction peaks representing the reduction of, respectively, elemental sulfur to long-chain polysulfides and long-chain poly-sulfides to short-chain polysulfides[75]. However, the two peaks were polarized from 2.32 V and 2.00 V in PEB-1 cells to, respectively, 2.29 V and 1.96 V in PVDF cells. This is consistent with our observations that PEB-1 possesses high intrinsic ionic conductivity, which improves electrode kinetics, whereas PVDF does not. It should be noted that an extra minor reduction peak appears in PEB-1 cells at ~2.47 V. This extra peak likely arises from the formation of polysulfides with pyrrolidinium cations instead of Li$^+$ cations, which increases the redox potential. As a comparison, PEB-1/Carbon Black does not show such peaks (Supplementary Fig. 1), signifying PEB-1 is not redox active by itself. For the anodic peaks, there is one major oxidation peak (2.35 V) with one shoulder peak (2.43 V).

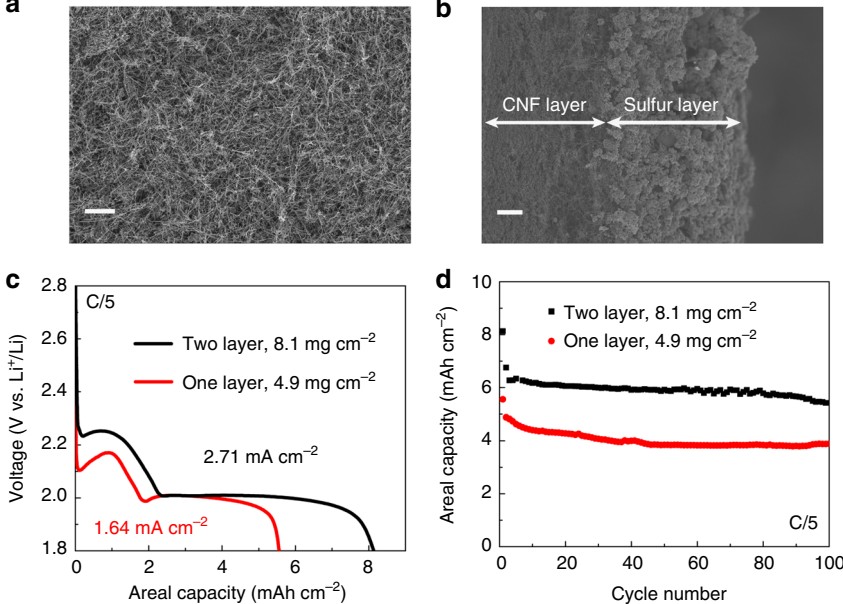

**Fig. 5** High-loading Li–S cells with CNF current collector. **a** SEM image of the top surface of CNF current collector (scale bar = 10 μm). **b** SEM image of the cross-section of the sulfur cathode with CNF current collector (scale bar = 20 μm). **c** The 1st cycle discharge voltage profiles of the cells with one-layer or two-layer highly loaded sulfur cathodes deposited onto CNF current collectors. **d** Cycling performance of the cells with one-layer or two-layer CNF current collector sulfur cathodes

Long-term CV scans highlighted the stability of the cells influenced by binders. In both cells, the reduction peak potentials of the 1st cycle are slightly lower than those that follow, signifying an activation process in the initial cycle. Beyond the 1st cycle, the CV profiles of PEB-1 cells overlap for ~200 cycles, indicating excellent stability for the sulfur cathode. The extra redox peaks from pyrrolidinium polysulfides around 2.47–2.50 V are reversible for 200 cycles, which means that the pyrrolidinium polysulfides are not only active but also quite reversible and stable over long-term cycling. Meanwhile, CV profiles of PVDF cells are only stable up to 50 cycles; a gradual degradation is observed afterward. At around 180 cycles, the redox peaks almost disappear, showcasing the poor long-term stability of PVDF cells.

The cell kinetics were further compared by testing rate capabilities from C/5 to 2 C (Fig. 4c, d). For PEB-1 cells, the discharge capacities are around 1244, 1051, 939, and 821 mAh g$^{-1}$ at C/5, C/2, C, and 2 C rates, which are considerably higher than the values of PVDF cells, which are around 878, 591, 486, and 249 mAh g$^{-1}$. The advantage of PEB-1 over PVDF binder is more obvious at higher current densities. This is owing to the high ionic conductivity of the binder, which contributes to high utilization of sulfur within the electrodes, especially at high rates when mass transport is the limiting step. PEB-1 binder enables the maximum utilization of active sulfur even in the depths of the mesoporous carbon.

In order to evaluate the long-term cycling stability, galvanostatic cycling of the Li–S cells with either PEB-1 or PVDF binder was carried out at a rate of C/5 (Fig. 4e). After 250 cycles, a high capacity of 731.1 mAh g$^{-1}$ was still retained for PEB-1 cells, which is owing to the polysulfide-trapping character of PEB-1. Meanwhile, PVDF cells underwent faster degradation, within 200 cycles; after ~220 cycles, abrupt failure of the PVDF cells occurred. As PVDF does not bind polysulfides strongly, more sulfur loss to the anode side can be expected with each cycle. Polysulfides react with lithium metal anode, as do the components in the electrolyte. In that PVDF cells failed fast, loss of both active sulfur and electrolyte are at fault. The evolution of polarization of the cells is evaluated by comparing the

discharge–charge voltage profiles of the 1st and 200th cycles of PVDF and PEB-1 cells (Supplementary Fig. 11). For PEB-1 cells, there is no significant increase in overpotential due to the good stability and integrity of the sulfur electrodes, which contributes to fewer side reactions on the lithium metal anode as well. As a comparison, cells with PVDF binder suffer from a large increase in overpotential.

For a more practical evaluation of the PEB-1 binder for commercial Li–S batteries, highly loaded sulfur cathodes configured with PEB-1 were also tested. One problem we faced with higher loading is that solution-processed composites tended to peel off Al-CCs after drying. To overcome this delamination issue, we replaced the Al-CC with a CNF-CC, which allowed us to increase the loading from 1.2 mg cm$^{-2}$ to ~ 4 mg cm$^{-2}$. As shown in Fig. 5a, the CNF-CC consists of interwoven nanofibres with abundant interstitial gaps. This highly porous structure is able to accommodate the volume change during the slurry drying process. After coating the CNF-CC with the sulfur composite, two distinct layers are observable (Fig. 5b). Notably, N-MC/S does not penetrate deep into the CNF web; instead, N-MC/S particles are adhered to the surface. By stacking two layers of sulfur-loaded CNF, a high loading of ~ 8 mg cm$^{-2}$ was achieved. The cycling behavior of the cells with a loading of 4.9 mg cm$^{-2}$ and 8.1 mg cm$^{-2}$ at a rate of C/5 is shown in Fig. 5c. With a loading of 4.9 mg cm$^{-2}$, the cell delivered a high initial areal capacity ~5.56 mAh cm$^{-2}$ (Fig. 5c), which slowly decreased to 3.89 mAh cm$^{-2}$ after 100 cycles (Fig. 5d). This amounts to an average 0.30% decrease in capacity per cycle. Even with an ultra-high loading of 8.1 mg cm$^{-2}$, the specific capacity decreased slightly from 8.13 mAh cm$^{-2}$ to 5.42 mAh cm$^{-2}$, whose average fade rate is 0.33% (i.e., slightly higher than the 4.9 mg cm$^{-2}$ cell). Compared with leading-edge sulfur cathodes with advanced binders (Supplementary Table 1)[1,7,73,74,76–82], our cells show high capacities at a fast rate with a high sulfur mass loading and long cycle life, which is attributed to the combined advantages of the large N-doped surface area of our N-MC and facile Li$^{+}$-ion transport in the electrode as aided by PEB-1. These results further demonstrate that the PEB-1 binder is efficient in

inhibiting the loss of active sulfur species, improving capacity retention.

## Discussion

Here we have laid the groundwork for understanding, both experimentally and theoretically, the molecular basis by which polyelectrolyte binders actively exert their influence on the diffusive transport of various ionic species encountered in the cycling of sulfur electrodes. Their role in this regard concerns both the facilitated transport of lithium ions throughout the electrode, which is key to attaining fast $S/Li_2S$ interconversion kinetics at high current densities, and restricted active material diffusion, which is critical in minimizing capacity fade at high sulfur loading. Specifically, we found that the hydrophobic and covalent character of higher order and electrolyte-soluble lithium polysulfides leads to preferential and strong electrostatic interactions with the cationic polymer backbone, which could be leveraged to prevent their diffusion from the cathode on cycling. From this bound state, these polysulfides could either be oxidized on the charge to solid sulfur, thereby preventing further diffusion, or easily reduced on the discharge to shorter oligomers. Furthermore, on reduction, the ionic character of lithium polysulfides increases as the oligomer length decreases along the discharge. We find that the energy holding those ionic polysulfides to the polymer decreases considerably, allowing the critical concentration of $Li_2S_4$ to be reached and the precipitation of $Li_2S_2/Li_2S$ to occur locally as desirable. More importantly, these functions enabled by the PEB-1 binder do not appear to be limited by areal sulfur loading, which is unusual. A likely explanation is that a significant fraction of the polysulfide trapping occurs at the interface of the electrolyte and the porous carbon host for sulfur-active materials. We also find that the implementation of mobile anions counterbalancing cationic residues along the polymer binder's backbone greatly improves the reaction kinetics for $S/Li_2S$ interconversion and lowers considerably the cell impedance, allowing the accessible capacity to remain high throughout long-term cycling, even with a high mass loading of 8.1 mg cm$^{-2}$, where PEB-1 cells deliver a specific capacity of 1004 mAh g$^{-1}$ at a moderate rate of C/5. Given that PEB-1 could be easily scaled to meet the demands for high-volume production, it may be a good choice for advanced Li–S battery manufacturing, as might other cationic polyelectrolytes with mobile anions (e.g., $PF_6^-$, $TfO^-$, $FSI^-$).

## Methods

**Materials**. Phenol (99 + %), NaOH (97 + %), conc. HCl (37%), poly(N,N-diallyl-N,N-dimethylammonium) chloride (PDDA-Cl, $M_n$ = 400–500 kg mol$^{-1}$, 20% w/w in H$_2$O), Ludox HS40 silica colloid (40% w/w in H$_2$O), 1,3-dioxlane (DOL, 99.8%), cyanamide (99%), 1,2-dimethoxymethane (DME, 99.5%), lithium nitrate (LiNO$_3$, 99.99%), and lithium bis(trifluoromethanesulfonimide) (LiTFSI, 99.95%) were obtained from Sigma-Aldrich. Formaldehyde solution (37% w/w in H$_2$O), lithium metal strip (0.75 mm thick, 99.9%), and sulfur (99.5%) were obtained from Alfa Aesar. Hydrofluoric acid (48%) was obtained from Acros Organics. Ethanol (88.5–92.5% v/v) was obtained from Macron Fine chemicals. CNF (>98%) was purchased from Sigma-Aldrich.

**Resol synthesis**. Phenol (12.0 g, 128 mmol) was heated in a round-bottom flask at 65 °C until molten, after which aqueous NaOH (2.50 g, 20% w/w in H$_2$O) was added to the flask dropwise[83]. Aqueous formaldehyde (21.0 g, 37% w/w in H$_2$O) was then added, and the mixture aged for an additional 50 min at 65 °C. The mixture was subsequently neutralized with aqueous HCl. Water was removed from the reaction mixture in vacuo to obtain the resol. Finally, an equal weight of ethanol was mixed with the resol to form the resol ethanol solution (50% w/w).

**N-MC and N-MC/S composite preparation**. To prepare the N-MC[84], the resol ethanol solution (1.0 g, 50% w/w) was mixed with cyanamide (0.50 g, 12 mmol) and HS40 silica colloid (3.0 ml) and sonicated for 10 min. Afterward, the transparent yellow solution was dried at 50 °C overnight under continuous stirring, thermo-polymerization at 100 °C for 24 h, and carbonization of the resulting monolith at

800 °C for 2 h under Ar (heating and cooling rate = 2 °C min$^{-1}$). To etch away the silica template, the black monolith was ground into powder and immersed in HF (20% w/w in H$_2$O) for 24 h. The particulates were isolated by filtration, and the filter cake washed with copious amounts of DI water. The N-MC product was subsequently dried at 50 °C overnight prior to use. SEM: see Supplementary Fig. 7a. TEM: see Supplementary Fig. 7b. BET: see Supplementary Fig. 7c and d. XPS: see Supplementary Fig. 8. For the preparation of N-MC/S composite, N-MC was initially mixed with pure sulfur (weight ratio = 2:8) using a mortar and pestle. The melt-infusion of sulfur into N-MC was then conducted at 155 °C for 12 h. TGA of N-MC/S: see Supplementary Fig. 9.

**PEB-1 synthesis**. PEB-1 was synthesized by anion metathesis. Briefly, PDDA-Cl (20.0 g, 20% w/w in H$_2$O, 10.0 µmol) was diluted with DI water (100 ml) prior to the addition of LiTFSI (8.52 g, 29.7 mmol) in DI water (10 ml). PEB-1 was collected as a colorless solid after vacuum filtration and drying in vacuo. Analytical characterization—i.e., $^1$H NMR: see Supplementary Fig. 12, FTIR: see Supplementary Fig. 13, TGA: see Supplementary Fig. 14, EA: see Table S2, etc—was in agreement with a previous synthesis[45].

**Characterization**. SEM was carried out using a Zeiss Gemini Ultra-55 analytical Field Emission Scanning Electron Microscope. TEM was carried out using a JEOL 2100 F at an acceleration voltage of 200 kV. UV-visible-spectra were collected with a Cary 5000 UV-Vis-NIR spectrophotometer. XPS measurements were performed using a Specs PHOIBOS 150 hemispherical energy analyzer using a mono-chromated Al Kα X-ray source. The load-lock of the analytical UHV system is connected directly to an Ar-filled glove box, enabling the loading of samples without any exposure to ambient atmosphere. Powder samples were mounted on carbon tape supported by Si substrates. Charge neutralization was carried out using a low energy flood gun (electron energy ≤ 5 eV), with the neutralization conditions optimized based on the degree of charging present for a given sample. Survey spectra were measured using a pass energy of 40 eV at a resolution of 0.2 eV/step and a total integration time of 0.2 s/point. Core level spectra were measured using a pass energy of 20 eV at a resolution of 0.05 eV/step and a total integration time of 0.5 s/point. Deconvolution was performed using CasaXPS software with a Shirley-type background and 70–30 Gaussian-Lorentzian peak shapes. Spectra were charge referenced using the position of aliphatic carbon in the C 1s peak at 284.8 eV. Sulfur K-edge X-ray absorption near edge structure spectra were collected at the Advanced Light Source beamline 10.3.2[85]. Scans were taken from 2410 to 2525 eV with an energy resolution of 0.25 eV. All spectra were calibrated to a gypsum reference standard. Data were acquired using an Amptek silicon drift detector at five spots on each sample and averaged together to increase the signal-to-noise ratio without inducing beam damage. Normalization and pre-edge background subtraction were performed using software provided at the beamline.

**Fabrication of sulfur cathodes**. A thin slurry was formed by mixing N-MC/S composite, binder (PEB-1 or PVDF), and Super P in NMP in a weight ratio of 7:1:2 and stirred overnight. The well-dispersed slurry was then coated onto an aluminum foil substrate by using a doctor blade. The coated electrodes were dried overnight at 50 °C under vacuum before being cut into circular disks with a diameter of 1.2 cm. The mass loading of sulfur in the sulfur electrodes was around 1.2 mg cm$^{-2}$. Alternatively, a CNF current collector was prepared by a vacuum-filtration process as reported previously[17]. The CNF current collector was cut into circular disks with a diameter of 1.2 cm with mass ~ 2 mg. N-MC/S and PEB-1 mixture slurry (weight ratio 9:1) was then drop cast on the CNF current collector and dried overnight at 50 °C under vacuum. Each CNF current collector contained a sulfur mass loading of 4–5 mg cm$^{-2}$.

**Electrochemical characterization**. Li–S batteries were tested with CR2032-type coin cells. The sulfur cathode and lithium metal anode were separated by a single Celgard 2400 separator. The electrolyte was made of 1.0 M LiTFSI and 0.2 M LiNO$_3$ dissolved in DOL/DME (1:1 v/v). The electrolyte:sulfur ratio (E:S) was ~ 10 mL$_E$ g$_S^{-1}$. Electrochemical experiments were carried out using a Biologic VMP3 potentiostat. The galvanostatic cycling tests at different C rates (1 C = 1675 mA h g$^{-1}$) were conducted within the voltage range of 1.8–2.8 V. Impedance data were recorded at open circuit voltage (OCV) in the frequency range of 1 MHz to 1 Hz with an AC voltage amplitude of 10 mV. CV measurements were conducted between 1.8–2.8 V at a scan rate of 0.1 mV s$^{-1}$. GITT for the cell discharge was conducted from OCV to 1.8 V at C/5 with 5 min discharge interval and 30 min delays. For potentiostatic electrodeposition, cells were equilibrated at 2.3 V to transform sulfur into long-chain polysulfides before driving the Li$_2$S electro-deposition at constant voltage: either 2.0 V or 1.9 V.

**Computational methods**. Detailed description of all computational methods can be found in the Supplementary Methods.

**Data availability**. The data sets generated and/or analyzed in this study are available from the corresponding author on reasonable request.

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

## Acknowledgements

This work was supported by the Joint Center for Energy Storage Research, an Energy Innovation Hub funded by the US Department of Energy, Office of Science, Office of Basic Energy Sciences. Portions of the work—including PEB-1 and N-MC synthesis, characterization, and some testing in Li–S cells—were carried out as a user project at the Molecular Foundry, which is supported by the Office of Science, Office of Basic Energy Sciences, of the US Department of Energy under contract no. DE-AC02-05CH11231. This research used resources of the National Energy Research Scientific Computing Center and the Advanced Light Source, DOE Office of Science User Facilities supported by the Office of Science of the US Department of Energy under that same contract. XPS measurements were completed at the Electrochemical Discovery Laboratory at Argonne National Laboratory. T. E. Williams is thanked for assistance with TEM.

## Author contributions

B.A.H. developed the concept and directed the experimental design. L.L. synthesized, characterized, and applied the binder in Li–S cells. T.A.P. and D.P. carried out all molecular dynamics (MD) simulations and free-energy calculations. J.G.C. conducted XPS measurements and analysis. L.L., B.A.H., F.Y.F, and Y.-M.C. designed and analyzed GITT and potentiostatic discharge experiments. S.M.M. performed XAS measurements and analysis. L.M. conducted TGA and FTIR analysis of PEB-1. The manuscript was written with contributions from all the authors. All authors have given approval to the final version of the manuscript.

## Additional information

**Competing interests:** The authors declare no competing financial interests.

