## [Peer Review File · Nature Communications]

Reviewers' comments:

Reviewer #1 (Remarks to the Author):

This paper presents the use of poly(DADMAC) TFSI poly(ionic liquid)s as binders for Lithium Sulfur batteries. Lithium-Sulfur batteries is one of the "hottest" technologies and the focus of the role of polymeric binders as presented in this paper is quite interesting for the battery community. In general the paper is very well written combining experimental and theoretical measurements. However, I do have however several important issues that need to be taken in to account:

1. The paper uses a poly(ionic liquid) as binder but does not include any references related to the previous use of poly(ionic liquid)s as binders in batteries. Important references to the pioneering works by Passerini and co workers and Antonietti and co-workers in the use of poly(ionic liquid)s as binders in lithium batteries should be included.

2. The paper does not make any mention to other active binders able to trap sulfides that were reported in the past such as the similar poly(DADMAC) chloride or poly(ethylene oxide) binders. Mentions to these previous works should be made in the text and the data added to Table S1.

3. One important role of the binder and the further performance of the electrode is the distribution of the active S particles and the porosity of it. The paper does not show a clear comparison between the PVDF used electrodes and the PEB-1. SEM of the electrodes and porosity measurements should be included. In fact the performance of the electrodes may be related to this...

4. Can the authors quantify the amount of sulfides trapped in the PEB-1?

5. If the binders makes that the concentration of LiTFSI increases, would this affect the performance of the battery? May this be a reason?

5. The article demonstrates the regulation of ion transport binder with only one particular type of polyelectrolyte (POLYDADMAC-TFSI). In order to make it a general statement more than one example should be given. Can this be extended to other type of polyelectrolytes asuch as POLYDADMAC-FSI or poly(vinylimidazolium)-TFSI ones?

Reviewer #2 (Remarks to the Author):

Helms and coworkers investigated the effect of the cationic polyelectrolyte, PEB-1, as an active binder for sulfur cathodes with high areal sulfur loadings (up to 8.1 mg cm^{-2}). Interestingly, PEB-1 can effectively undergo anion metathesis with lithium polysulfides, which can prevent polysulfide dissolution from the sulfur host and also simultaneously increase the ionic conductivity. The authors clearly demonstrated this fact by comparing the electrochemical performance of PEB-1 with that of PvDF binder. The authors also demonstrated the decent cycling performance at a high loading of sulfur. Overall, this is an interesting article and should be suitable for publication once the below noted questions have been adressed.

While this reviewer notes the improvement at high sulfur loadings, even PvDF performed better under these conditions, therefore, in order to exclusively demonstrate the effect of PEB-1 at high loadings, the authors should carry out the same experiments using Al as a current collector (to eliminate the effect of carbon nanofiber). Accordingly, they should also specify the current collector used in their comparison table, Table S1. In addition, it should also be specified whether the discharge capacity is calculated based on total mass or sulfur mass only. Very recent binder article published in EES (Energy & Environ. Sci., 2017, 10, 750-755) should be also cited and compared.

During lithiation, lithium polysulfides are generated, and some of them are trapped by PEB-1 (considering that only 10 wt% on the binder is added) in the form of pyrrolidinium polysulfide. Where does the excess polysulfides go? it is not realistic to assume that there will be perfect

stoichiometry match between cationic sites and Li-polysulfides? In addition, how does the electrochemical reaction of the pyrrolidinium polysulfide occur during delithiation? Are those pyrrolidinium polysulfides still active? If so, the operating voltage would be different from that of lithium polysulfide. What voltage is it? The evidence for this mechanism should be given.

On page 6 (line 119), the authors argue that PEB-1 cells showed higher peak current density than that of PVDF in Fig. 2b. However, the current intensity depends on the loading amount of the active material. It will be more appropriate to change the unit of current from mA to mA/g.

it is necessary to test without LiTFSI / LiNO₃ to show real Lipolysulfides adsorption property of PEB-1 as these pre-dissolved slats are known to decrease the solubility of Lipolysulfides in the electrolyte mixture.

In the XAS plot, where is the peak at 2497 for PEB-1? Why do the sulfonyl peaks remain even after TFSI⁻ in PEB-1 exchanged with polysulfide?

As reported by Armand and Chen (Nat. Commun., 2013, 4, 1481), concentration of LiTFSI in electrolyte has significant effect on Li-S battery performance by reducing the solubility of polysulfides. In addition, there are also reports that electrolyte decomposition lead to severe decomposition of LiTFSI to Li₃N, LiF, or sulfonyl compounds (Nat. Commun., 2013, 4, 1481, Journal of The Electrochemical Society, 2009, 156, A694, 2012, 159, A1816-A1821), and these are co-deposited in cathodes causing capacity decay. If the PEB-1 expedite lithium ion transport and increase the overall cell performance as author mentioned, lithium cycling efficiency experiment and cell cycling data without LiTFSI would be necessary.

Graph assignment in Figure S9 is wrong. Black line should be 'PEB-1'. Some figure numbers are also not matched. On page 14, the supporting figure number (Fig. S9, line 277) indicating TGA is needed to be corrected to Fig. S10. On page 18 (line 347), the figure number (Fig. 5b) indicating the cycling behavior should be Fig. 5c.

Some important references on the covalent stabilization of elemental sulfur should also be included; Angew. Chem. Int. Ed., 2016, 55, 3106–3111. ACS Energy Lett., 2016, 1, 566–572. Nat. Chem. 2013, 5, 518–524. Nat. Commun. 2015, 6, 7278.

RESPONSE TO REVIEWER'S COMMENTS

REVIEWER 1:

General Comments: This paper presents the use of poly(DADMA) TFSI poly(ionic liquid)s as binders for Lithium Sulfur batteries. Lithium-Sulfur batteries is one of the "hottest" technologies and the focus of the role of polymeric binders as presented in this paper is quite interesting for the battery community. In general the paper is very well written combining experimental and theoretical measurements. However, I do have however several important issues that need to be taken in to account:

Answer to general comments: We thank the reviewer for the positive comments. No general comments to address.

Comment 1: The paper uses a poly(ionic liquid) as binder but does not include any references related to the previous use of poly(ionic liquid)s as binders in batteries. Important references to the pioneering works by Passerini and coworkers and Antonietti and co-workers in the use of poly(ionic liquid)s as binders in lithium batteries should be included.

Answer to comment 1: We thank the reviewer for the suggestions. We have now added the following references on poly(ionic liquid)s as binders in batteries by Passerini et al. and Antonietti et al. in the manuscript:

40. Gebresilassie Eshetu, G., Armand, M., Scrosati, B., & Passerini, S. Energy Storage Materials Synthesized from Ionic Liquids. *Angew. Chem. Int. Ed.* **53**, 13342–13359 (2014).

41. Osada, I., de Vries, H., Scrosati, B. & Passerini S. Ionic-Liquid-Based Polymer Electrolytes for Battery Applications. *Angew. Chem. Int. Ed.* **55**, 500–513 (2016).

42. Yuan, J., Mecerreyes, D. & Antonietti, M. Poly(ionic liquid)s: An update. *Prog. Polym. Sci.* **38**, 1009–1036 (2013).

In order to make it clear, we added the following information on Page 4–5 of the manuscript:

“Our results are complementary to advances in Li-ion battery technology development using polyelectrolyte binders (e.g., poly(ionic liquid)s), which yielded cells with high specific capacity and excellent long-term electrochemical stability when compared to PVDF binder⁴⁰⁻⁴². The emerging perspective is that the design space for polyelectrolyte binders is superior, allowing for broad tunability of

electrochemical stability (both anodic and cathodic), energetic barriers to Li⁺ desolvation and transport, and adhesion.”

Comment 2: *The paper does not make any mention to other active binders able to trap sulfides that were reported in the past such as the similar poly(DADMAC) chloride or poly(ethylene oxide) binders. Mentions to these previous works should be made in the text and the data added to Table S1.*

Answer to comment 2: We thank the reviewer for the suggestion. We have now added the following references in both the text and Table S1. In doing so, the superior performance of PEB-1 is again evident and noteworthy.

72. Zhang, S. S. Binder based on polyelectrolyte for high capacity density lithium/sulfur battery. *J. Electrochem. Soc.* **159**, A1226–A1229 (2012). (poly(DADMAC) chloride)

73. Lacey, M. J., Jeschull, F., Edstrom, K. & Brandell, D. Why PEO as a binder or polymer coating increases capacity in the Li–S system. *Chem. Commun.* **49**, 8531–8533 (2013). (poly(ethylene oxide))

74. Cheon, S.-E., et al. Structural factors of sulfur cathodes with poly(ethylene oxide) binder for performance of rechargeable lithium–sulfur batteries. *J. Electrochem. Soc.* **149**, A1437–A1441 (2002). (poly(ethylene oxide))

To make it clear, we added the following information on Page 14–15 of the manuscript:

“PEB-1 may also find more practical use than other cationic binders for Li–S cells, e.g., poly(acrylamide-co-diallyldimethylammonium chloride) (PAMC), which corrodes Al current collectors⁷². By obviating the use of corrosive Cl[–] counterions, PEB-1 is a preferred embodiment. It may also be the case that PEB-1 influences the electrolyte system within the electrode’s pores, altering the chemistry and solvated structures of polysulfides and the working ion in the pores. This has been suggested for binders like poly(ethylene oxide) (PEO)^{73,74}, which delays the passivation of the cathode at end of discharge rather than a trapping capacity for polysulfides⁷³.”

Comment 3: *One important role of the binder and the further performance of the electrode is the distribution of the active S particles and the porosity of it. The paper does not show a clear comparison between the PVDF used electrodes and the PEB-1. SEM of the electrodes and porosity measurements should be included. In fact the performance of the electrodes may be related to this.*

Answer to comment 3: We thank the reviewer for the suggestion. We have now added the SEM images of the PVDF and PEB-1 electrodes as Supplementary Fig. S10.

As we can tell from the SEM images, there are no apparent differences between these two electrodes, since the dominant feature is the N-MC/S composite covered by Super P particles. Binder only occupies 10% of the total weight, which is amorphous and does not possess discernable morphological features.

For the porosity measurements, we carried out BET analysis on N-MC and N-MC/S (Supplementary Fig. S7). By comparing isotherms and pore size distribution of N-MC and N-MC/S, all mesopores in N-MC/S are filled by sulfur and the surface area was dropped to $5.2 \text{ m}^2 \text{ g}^{-1}$. Thus, it will make no significant difference when a small amount of different binders are added into the N-MC/S samples.

To make it clear, we added the following information on page 16 in the manuscript:

“SEM images of PEB-1 and PVDF electrodes (Supplementary Fig. S10) show few macroscopic structural differences with respect to the distribution of N-MC/S composite and Super P particles (neither binder is directly observable). This is not too surprising as the binder loading is similar for both electrode formulations, and the surface area of N-MC/S is small ($5.2 \text{ m}^2 \text{ g}^{-1}$, see Supplementary Fig. S7).”

Comment 4: *Can the authors quantify the amount of sulfides trapped in the PEB-1?*

Answer to comment 4: For the polysulfide absorption experiment shown in Supplementary Fig. S2a, the amount of sulfides trapped in the PEB-1 is 96% of the total amount of sulfides in the solution, which was quantified by UV-vis. It is not surprising because when we prepared the PEB-1/Li₂S₆ sample, the ratio between the monomer in PEB-1 and Li₂S₆ is 5 to 1. This means that nearly all polysulfides could be bonded to PEB-1 when PEB-1 is in excess.

In a Li-S coin cell, it is almost impossible to quantify the amount of polysulfides bound to PEB-1 through a quantitative in situ experiment; furthermore, there is only 0.242 mg PEB-1 in each cell. At the same time, a post-mortem analysis would require clean isolation of the composite, probably from many cells, without contamination from carbon and electrolyte, which absorb or contain sulfur as well.

Nevertheless, we can perform the following thought-experiment. In a Li-S coin cell, the amount of polysulfides absorbed by PEB-1 can be estimated as follows. On a molar basis, there are significantly more polysulfides than cationic repeat units along PEB-1 chains. Polysulfides will have to make a choice: either be bound to lithium or be bound to the cationic polymer moieties. If only one end of each polysulfide is attached to each pyrrolidinium cation, with the other end coordinated to lithium, up to 10% of polysulfides (in the form of Li₂S₈) in the system could be associated with the binder through electrostatic interactions. If both ends of polysulfides are attached to pyrrolidinium cations, up to 5% polysulfides (in the form of Li₂S₈) could be bound. Based on our MD simulations and prior work regarding the prevalence of dianionic

polysulfides, we reason the former case is closer to reality. Additionally, the intrinsic polysulfide solubility in electrolyte and the extrinsic total electrolyte volume will determine the exact quantity of polysulfides in the system at a specific state of charge, allowing the presence of a cationic polysulfide adsorbent to alter the equilibrium polysulfide solubility to a small extent. With this in mind, 5% polysulfide binding capacity should be considered an upper bound. However, if this trapping occurs at the orifice of a mesopore, it effectively gates the pore from allowing polysulfides to diffuse further, and thus its impact on capacity retention is formidable.

Comment 5: *If the binders makes that the concentration of LiTFSI increases, would this affect the performance of the battery? May this be a reason?*

Answer to comment 5: The rate of reaction is directly proportional to the local concentration of [LiTFSI]. As lithium polysulfides are generated in the electrode, within the confines of the mesopores of N-MC hosts, LiTFSI is consumed and depleted near the reaction front. In the absence of PEB-1, LiTFSI diffusion from the electrolyte into the mesopore is rate-limiting. In the presence of PEB-1, the generation of lithium polysulfides leads to an ion metathesis that transiently increases the local concentration of LiTFSI. The positive effect of this is, in the end, transient: once ion metathesis is complete, no additional LiTFSI can be generated. The lasting effect, however, is the low overall impedance of the electrode with PEB-1 in place. The kinetic consequences of that are profound and delineated as such in the course of this manuscript.

Comment 6: *The article demonstrates the regulation of ion transport binder with only one particular type of polyelectrolyte (POLYDADMA-TFSI). In order to make it a general statement more than one example should be given. Can this be extended to other type of polyelectrolytes such as POLYDADMA-FSI or poly(vinylimidazolium)-TFSI ones?*

Answer to comment 6: We thank the reviewer for this suggestion. We fully expect our results to translate to other cationic polymers (PDADMA, poly(vinylimidazolium), poly(vinyltriazoles), polystyrenes with cationic pendants like cyclopropeniums, quaternary amines, quaternary phosphines, etc.) with mobile anions (TFSI⁻, PF₆⁻, TfO⁻, FSI⁻, etc.). For example, in Ref. 45 of the manuscript, PDADMA with different anions (TFSI⁻, PF₆⁻, BF₄⁻, DBSA) were synthesized as a polymer electrolyte for Li-ion batteries, which could in principle be applied as binder materials in Li-S batteries. But care should be taken that the anions should be stable with sulfur species upon cell operation.

To make it clear, we added the following information on page 21 of the manuscript:

“...other cationic polyelectrolytes with mobile anions (e.g., PF₆⁻, TfO⁻, FSI⁻).”

REVIEWER 2:

General Comments: Helms and coworkers investigated the effect of the cationic polyelectrolyte, PEB-1, as an active binder for sulfur cathodes with high areal sulfur loadings (up to 8.1 mg cm^{-2}). Interestingly, PEB-1 can effectively undergo anion metathesis with lithium polysulfides, which can prevent polysulfide dissolution from the sulfur host and also simultaneously increase the ionic conductivity. The authors clearly demonstrated this fact by comparing the electrochemical performance of PEB-1 with that of PvDF binder. The authors also demonstrated the decent cycling performance at a high loading of sulfur. Overall, this is an interesting article and should be suitable for publication once the below noted questions have been addressed.

Answer to comments: We thank the reviewer for the positive comments. No general comments to address.

Comment 1: While this reviewer notes the improvement at high sulfur loadings, even PvDF performed better under these conditions, therefore, in order to exclusively demonstrate the effect of PEB-1 at high loadings, the authors should carry out the same experiments using Al as a current collector (to eliminate the effect of carbon nanofiber). Accordingly, they should also specify the current collector used in their comparison table, Table S1. In addition, it should also be specified whether the discharge capacity is calculated based on total mass or sulfur mass only. Very recent binder article published in EES (Energy & Environ. Sci., 2017, 10, 750-755) should be also cited and compared.

Answer to comment 1: We thank the reviewer for the suggestions. We have now made the following revisions to address these issues.

1. We have tried to achieve a higher mass loading ($> 2 \text{ mg cm}^{-2}$) on pure Al (aluminum) current collector, but failed due to delamination of the composite. This is a common problem in Li-S battery research. As we can see from the 12 papers in Table S1, an Al current collector can only achieve a mass loading $< 2 \text{ mg cm}^{-2}$. To achieve a higher loading, special current collectors are needed: carbon-coated Al foil, nickel foam, or carbon nanofiber paper.
2. We have now added a column in Table S1, specifying the current collectors used in each paper. We can find that special current collectors other than pure Al foil are usually needed for sulfur electrodes with high areal capacity.
3. The specific capacity is calculated based on sulfur. In order to make it clear, we changed unit of discharge capacity in Table S1 from “mA h g⁻¹” to “mA h g⁻¹ sulfur)

On page 20 of the manuscript, we changed “our cells show higher capacities with a higher loading” to “our cells show high capacities at a fast rate with a high sulfur mass loading and long cycle life”.

4. We have now cited the recent binder paper on EES (*Energy & Environ. Sci.*, 2017, 10, 750-755) as Ref. 82 in the manuscript. The EES paper has also been added into Table S1 for comparison. As we can see from Ref. 82 in Table S1, although it reports a higher sulfur mass loading of 19.8 mg cm^{-2} , the rate is much lower (C/40) and cycle life is much shorter (5 cycles).

Comment 2: *During lithiation, lithium polysulfides are generated, and some of them are trapped by PEB-1 (considering that only 10 wt% on the binder is added) in the form of pyrrolidinium polysulfide. Where does the excess polysulfides go? it is not realistic to assume that there will be perfect stoichiometry match between cationic sites and Li-polysulfides? In addition, how does the electrochemical reaction of the pyrrolidinium polysulfide occur during delithiation? Are those pyrrolidinium polysulfides still active? If so, the operating voltage would be different from that of lithium polysulfide. What voltage is it? The evidence for this mechanism should be given.*

Answer to comment 2: We thank the reviewer for their comments on the working principle of our polyelectrolyte binder. Here are detailed explanations for each question:

1. *Where does the excess polysulfides go? it is not realistic to assume that there will be perfect stoichiometry match between cationic sites and Li-polysulfides?*

During discharge, lithium polysulfides are generated. As shown in Fig. 1a of our manuscript, owing to its architecture, N-MC retains most sulfur inside the carbon particles, which have a high internal to external surface area ratio. Nitrogen-doping promotes the chemical adsorption between sulfur atoms and N-MC, which is demonstrated in Ref.13 of the manuscript. In this way, polysulfides near the orifice of a mesopore are most subject to diffusion and loss. However, when N-MC/S is coated with PEB-1 during electrode processing, polysulfides diffusion can be effectively managed.

According to our calculation, a maximum of ~ 10% polysulfides (in the form of Li_2S_8) in the cathode can be bonded by PEB-1, which should be enough considering the high internal to external surface ratio of N-MC trapping most polysulfides inside.

To make it clear, we added the following information on page 15 of our manuscript:

“Owing to its architecture, N-MC retains most sulfur inside the carbon particles, which have a high internal to external surface area ratio. Nitrogen-doping promotes

the chemical adsorption between sulfur atoms and N-MC¹³. In this way, polysulfides near the orifice of a mesopore are most subject to diffusion and loss. However, when N-MC/S is coated with PEB-1 during electrode processing, polysulfides diffusion can be effectively managed. Toward that end...”

2. In addition, how does the electrochemical reaction of the pyrrolidinium polysulfide occur during delithiation? Are those pyrrolidinium polysulfides still active? If so, the operating voltage would be different from that of lithium polysulfide. What voltage is it?

During delithiation or charge process, the short-chain pyrrolidinium polysulfides will be oxidized to form long-chain pyrrolidinium polysulfides until S₈ is formed and pop off the pyrrolidinium anchor points. The pyrrolidinium polysulfides are still active. The operating voltage of pyrrolidinium polysulfides is a little bit higher than lithium polysulfides. As shown in Fig. 2b, a pair of small redox peaks appears around 2.47 V and 2.50 V, which is due to the formation of pyrrolidinium polysulfides instead of lithium polysulfides. And from Fig. 4a, we can see that the redox pair is stable for 200 scan cycles, which means the pyrrolidinium polysulfides are active and reversible.

In order to make it clear in the manuscript, we add the following information on page 17 of the manuscript:

“The extra redox peaks from pyrrolidinium polysulfides around 2.47–2.50 V are reversible for 200 cycles, which means that the pyrrolidinium polysulfides are not only active but also quite reversible and stable over long-term cycling.”

Comment 3: *On page 6 (line 119), the authors argue that PEB-1 cells showed higher peak current density than that of PVDF in Fig. 2b. However, the current intensity depends on the loading amount of the active material. It will be more appropriate to change the unit of current from mA to mA/g.*

Answer to comment 3: We thank the reviewer for the suggestion. We have now changed the unit of current from mA to mA/g in Fig. 2b.

Comment 4: *it is necessary to test without LiTFSI / LiNO₃ to show real Li polysulfides adsorption property of PEB-1 as these pre-dissolved salts are known to decrease the solubility of Li polysulfides in the electrolyte mixture.*

Answer to comment 4: We thank the reviewer for the suggestion. We prepared 10 mM Li₂S₆ solution without LiTFSI/LiNO₃ and did the same immersion test. The color of the solution changes a little bit due to the removal of salts in the electrolyte. The PEB-1 binder became red after immersion with the color of the filtrate became much lighter afterwards. The picture describing the phenomenon has been added as

Supplementary Fig. S2b. This continues to support our primary claim that PEB-1 absorbs polysulfides, with or without LiTFSI / LiNO₃.

To make it clear, we added the following information on page 11 of our manuscript:

“Similar phenomenon could be observed without LiTFSI and LiNO₃ in the electrolyte (Supplementary Fig. S2b), illustrating the strong absorption behavior of PEB-1, even without supporting salts.”

Comment 5: *In the XAS plot, where is the peak at 2497 for PEB-1? Why do the sulfonyl peaks remain even after TFSI- in PEB-1 exchanged with polysulfide?*

Answer to comment 5: We are sorry that the peak associated with PEB-1 at 2497 eV is absent because the spectra in Fig. 3c and d are cut off at 2490 eV. We limited our scan range to 2490 eV during the experimental acquisition on the synchrotron. The two peaks around 2481 and 2486 eV, nevertheless, adequately represent the sulfonyl group in PEB-1 with distinction.

When we prepared the PEB-1/Li₂S₆ sample, the ratio between the monomer in PEB-1 and Li₂S₆ is 5 to 1, which means there will be at least 3/5 of PEB-1 left unreacted. Since PEB-1 is particulate and not soluble in the electrolyte, native (i.e., TFSI-containing) PEB-1 persists at the core while PEB-1/Li₂S₆ composite forms at the surface. This outer layer is readily penetrated by the high-energy XAS beam, and thus both compositions are present in the final spectrum. That's why we still see sulfonyl peaks in the XAS spectra.

To make it clear, on page 13 of the manuscript, we changed “PEB-1 shows three major peaks around 2481, 2486, and 2497 eV...” to “PEB-1 shows two major peaks around 2481 and 2486 eV...”

Comment 6: *As reported by Armand and Chen (Nat. Commun., 2013, 4, 1481), concentration of LiTFSI in electrolyte has significant effect on Li-S battery performance by reducing the solubility of polysulfides. In addition, there are also reports that electrolyte decomposition lead to severe decomposition of LiTFSI to Li₃N, LiF, or sulfonyl compounds (Nat. Commun., 2013, 4, 1481, Journal of The Electrochemical Society, 2009, 156, A694, 2012, 159, A1816-A1821), and these are co-deposited in cathodes causing capacity decay. If the PEB-1 expedite lithium ion transport and increase the overall cell performance as author mentioned, lithium cycling efficiency experiment and cell cycling data without LiTFSI would be necessary.*

Answer to comment 6: We thank the reviewer for the suggestion. We have now included the three references and related discussions into our manuscript (Ref. 28, 53, and 54). But cell cycling without LiTFSI is not possible due to the following reason.

Anion metathesis generating LiTFSI only happens at the interface between PEB-1 and polysulfides. It changes the concentration of LiTFSI at the reaction front in a very small region. In the bulk of the cell, salts are still needed to act as the charge carrier. According to our calculation, 10 wt% PEB-1 in sulfur cathodes produces 4.77×10^{-7} mol LiTFSI, which leads to an overall concentration of 0.035 mol/L in the electrolyte (if diluted into all the electrolyte in the cell) and is too little to power a normal Li-S battery. But if the produced LiTFSI is localized in a small region, it will be enough to improve reaction kinetics, transport properties, and reduce the solubility of polysulfides at the reaction front. Thus, the benefit of generating LiTFSI only happens in a small interfacial region. Supporting salts are still needed in the bulk electrolyte as charge carriers. A Li-S cell without LiTFSI will not work properly due to the lack of charge carriers in the bulk electrolyte. No meaningful information could be obtained without LiTFSI supporting salts (typically 1.0 M).

Moreover, considering the improved cell kinetics with PEB-1 binder compared with PVDF binder, we believe that generating LiTFSI should play a positive role in the cell performance.

To make it clear, we added the following references in the manuscript:

28. Suo, L., Hu, Y.-S., Li, H., Armand, M. & Chen, L. A new class of solvent-in-salt electrolyte for high-energy rechargeable metallic lithium batteries. *Nat. Commun.* **4**, 1481 (2013).

53. Aurbach, D., Pollak, E., Elazari, R., Salitra, G., Kelley, C. S. & Affinito, J. On the surface chemical aspects of very high energy density, rechargeable Li-sulfur batteries. *J. Electrochem. Soc.* **156**, A694-A702 (2009).

54. Diao, Y., Xie, K., Xiong, S. & Hong, X. Insights into Li-S battery cathode capacity fading mechanisms: Irreversible oxidation of active mass during cycling. *J. Electrochem. Soc.* **159**, A1816-A1821 (2012).

We also added the following information on page 7–8 of the manuscript:

“There is some evidence that LiTFSI oxidizes sulfur compounds to higher oxidation states, which would contribute to capacity fade^{53,54}. However, based on the superior electrochemical performance of PEB-1 cells, this effect does not appear to factor strongly here²⁸.”

Comment 7: Graph assignment in Figure S9 is wrong. Black line should be ‘PEB-1’. Some figure numbers are also not matched. On page 14, the supporting figure number (Fig. S9, line 277) indicating TGA is needed to be corrected to Fig. S10. On page 18 (line 347), the figure number (Fig. 5b) indicating the cycling behavior should be Fig.

5c.

Answer to comment 7: We thank the reviewer for pointing out the errors. We have now made the following changes:

On page 15 of the Supplementary Information, the assignment of the black line in Fig. S11 has been corrected (Due to revisions in the revised version, the figure number is changed from Fig. S9 to Fig. S11).

On page 15 of the manuscript, the description of TGA has been corrected to Fig. S9 (Due to revisions in the revised version, the figure number of TGA is changed from Fig. S10 to Fig. S9)

On page 19 of the manuscript, the description of Fig 5b has been corrected to Fig. 5c.

Comment 8: *Some important references on the covalent stabilization of elemental sulfur should also be included: Angew. Chem. Int. Ed., 2016, 55, 3106–3111. ACS Energy Lett., 2016, 1, 566–572. Nat. Chem. 2013, 5, 518–524. Nat. Commun. 2015, 6, 7278.*

Answer to comment 8: We thank the reviewer for the suggestion. We have now added the following references in the manuscript:

8. Talapaneni, S. N., Hwang, T. H., Je, S. H., Buyukcakir, O., Choi, J. W. & Coskun, A. Elemental-sulfur-mediated facile synthesis of a covalent triazine framework for high-performance lithium–sulfur batteries. *Angew. Chem. Int. Ed.* **55**, 3106-3111 (2016).

9. Je, S. H., *et al.* Rational sulfur cathode design for lithium–sulfur batteries: Sulfur-embedded benzoxazine polymers. *ACS Energy Lett.* **1**, 566-572 (2016).

10. Chung, W. J., *et al.* The use of elemental sulfur as an alternative feedstock for polymeric materials. *Nat. Chem.* **5**, 518-524 (2013).

11. Kim, H., Lee, J., Ahn, H., Kim, O. & Park, M. J. Synthesis of three-dimensionally interconnected sulfur-rich polymers for cathode materials of high-rate lithium–sulfur batteries. *Nat. Commun.* **6**, 7278 (2015).

We also added the following information on page 4 of the manuscript:

“...including sulfur-rich polymers⁸⁻¹¹...”

Reviewers' comments:

REVIEWERS' COMMENTS:

Reviewer #1 (Remarks to the Author):

The authors have answered and taking into account all my previous comments. I recommend the paper to be accepted.

Reviewer #2 (Remarks to the Author):

As the authors satisfactorily addressed my original comments, I recommend the publication of this article in its current form.